# Determination of Phenolic Compounds by Capillary Zone Electrophoresis–Mass Spectrometry

**DOI:** 10.3390/molecules27144540

**Published:** 2022-07-16

**Authors:** Ruben Szabo, Attila Gaspar

**Affiliations:** Department of Inorganic and Analytical Chemistry, University of Debrecen, Egyetem ter 1, 4032 Debrecen, Hungary; szabodavidruben@gmail.com

**Keywords:** phenolic compounds, capillary zone electrophoresis, mass spectrometry, honey

## Abstract

A CZE-MS method was developed for the determination of several phenolic compounds (phenolic acids, flavonoids). Since the analysis of these components necessitates the application of basic conditions for CZE separation and negative ionization mode for MS detection, the simplest choice was to use 0.5 M NH_4_OH and IPA:water (1:1 *v*/*v*%) as the background electrolyte and sheath liquid, respectively. The LOD values ranged between 0.004–1.9 mg/L showing that there are relatively large differences in the ionization (and chemical) features of these compounds. The precision data were better than 0.75 RSD% for migration times and were between 5–8 RSD% for peak areas. In order to test the applicability of the developed method, a honey sample was analyzed.

## 1. Introduction

Phenolic compounds are molecules including several bioactive properties, thereby offering excellent nutritional and health-protective effects owing to their antioxidant features. These compounds having an aromatic ring with one or more hydroxyl groups can be classified to phenols, phenolic acids, flavonoids, tannins or lignans. The organoleptic properties (taste, color) of certain foodstuff (e.g., berries, coffee, wine, honey) are mainly determined by these phenolic compounds. More than 8000 (poly)phenolic compounds have been detected in natural substances. There are several constitutional isomers and a large part of them (mainly the flavonoids) occur as aglycone, glycosides or methylated derivatives [1].

The well-known method for the quantitative determination of total phenolic content is the Folin–Ciocalteu colorimetric method, which, however, cannot provide any information about the individual phenolic analytes. In order to distinguish the large number of phenolic compounds, the combination of a high efficiency separation method with high detection sensitivity is strongly advised. Such analyses demand the utilization of MS detection, since MS allows us to gain information about both the molecular weight and the structure of a component. Following these requirements, many works can be found in the literature about the determination of phenolic compounds by GC-MS [2,3,4] or LC-MS [5,6,7]. However, GC analysis generally necessitates the derivatization of phenolic acids due to their low volatility and the high temperature applied can damage the analyte components [8]. CE-MS is also considered a powerful hyphenated method, however, its utilization for the analysis of phenolic compounds is relatively rarely demonstrated in the literature [7,9,10].

CE is often applied with a UV detector, which is suitable for the sensitive detection of chromophoric phenolic compounds. CE is a good alternative and complement to chromatographic methods owing to its minimal sample/reagent volume requirement and the ease of sample handling (derivatization is not necessary). In addition, CE showed its superiority in analyzing complex matrices (e.g., food, agricultural or clinical samples). In the last 30 years, several works have been published about the CE analysis of phenolic compounds using UV detection, where the CE technique applied was CZE, exclusively [8,11,12,13,14,15]. The hyphenation of CE with MS detection through electrospray ionization enabled the analysis of thermally labile, non-volatile and polar components like phenolic compounds. The majority of CE-MS work in this field was aimed at the determination of phenolic compounds in food or plant samples [10,16,17,18,19,20,21,22]. Arráez-Román et al. [20] analyzed mostly flavonoids and they expected that positive polarity MS detection is less selective but more sensitive compared to the applied negative polarity detection. They utilized a complex extraction/preconcentration procedure including the mixing of the sample with Amberlite particles in solution, then sugars and other polar constituents of honey were removed and the phenolic compounds were eluted. Afterwards, the methanolic extract was concentrated to dryness under reduced pressure in a rotary evaporator at 50 °C.

Since there are several constitutional isomers of phenolic compounds, which cannot be distinguished even by high resolution MS (only with tandem MS studies), the analysis of these compounds in complex matrices is a challenge. The aim of this work was to develop a CE–ESI-MS method for the determination of several phenolic compounds possibly occurring in natural samples. Besides flavonoids, we tried to cover a wider range of compounds, including phenolic acids and a non-aromatic compound, camphoric acid, as well (Figure 1). In an effort to apply the simplest possible conditions for the analysis, the background electrolyte (BGE) of choice was NH_4_OH solution, which yielded good separation and ensured the basic pH for negative ion MS detection mode. In order to test the applicability of the method developed, a honey sample was analyzed.

## 2. Results and Discussion

### 2.1. The Optimization of the CZE Method

CZE is applicable for the separation of ions having different charge-to-size ratios. Phenolic acids are fully dissociated components with moderate basic electrolytes like tetraborate, because their pKa values are between 4–5. However, a much higher pH is necessary for the ionization of non-acidic phenols (pKa of phenol and its simple derivatives is around 10). In order to separate these components with CZE-MS, basic BGEs are typically used, containing ammonium acetate with ammonium hydroxide or triethylamine (TEA) at pH = 10 [13,18,20] or even pH = 11 [23]. Although TEA is relatively often applied in MS studies, its use is not advantageous due to its persistent memory effect when positive mode MS measurements are carried out [24]. Provided that a highly basic solution (pH > 11) is necessary for the separation that is also compatible with CZE-MS, the most straightforward choice is the use of NH_4_OH solution. Pure ammonia solution is acceptable as BGE for CZE and also preferred by MS detection (negative ionization mode). The application of pure ammonia solution yields simple mass spectra, leading to sensitive detection. The only concern when using pure ammonia solution as BGE is the lack of buffer capacity and the volatility of ammonia; however, these drawbacks can be remedied by the frequent replenishment of the BGE (which can be automatically accomplished in commercial CE instruments) from/in well-closed containers. It should also be noted that in some works 8–15% organic modifier like acetonitrile [15] was used, but the resolution was not improved for all compounds and the analysis time was prolonged.

For phenolic compounds the negative ionization mode is preferred due to its better selectivity, although the sensitivities of the measurements are moderate compared to the usual positive mode MS analyses [20]. The application of sheath liquid is necessary in cases when conventional CE-MS setups are utilized; however, its composition and flow rate should be optimized to minimize analyte dilution and to maintain the electrospray stability (i.e., to maximize detection sensitivity). In our work, sheath liquid consisted of IPA:water (50:50 *v*/*v*), with no added electrolyte. Although alkaline sheath liquid is used for negative ionization mode in general, the high pH of the BGE ensures the basic condition for the ESI even in the solution obtained after the mixing of the CE effluent (BGE) with the sheath liquid.

In basic BGE the majority of the 15 phenolic compounds studied could be well separated and a considerable difference in separation efficiency was not observed in the pH range between 9–12. Using 0.5 M NH_4_OH (pH = 11.4) solution as BGE, co-migrations were found only in few cases, but even these components could be easily distinguished recording the proper extracted ion electropherogram (EIE). In neutral BGE, fewer components (11 from the 15) could be detected because of the lower ionization degree of the non-acidic components and the poorer detection sensitivity, however, caffeic acid showed an extremely high signal intensity compared to the other components (Figure 2). Obviously, the migration order was largely changed between pH = 7 and pH = 11.4, since the compounds studied show large differences in pK values.

The necessary application of sheath liquid for CE-ESI/MS offers the possibility to use positive MS mode even when the electrophoretic separation is carried out in basic conditions, since the sample solution can be largely acidified if an acid (e.g., 0.1% formic acid) is added to the sheath liquid. In Figure 3, the CZE-MS electropherograms obtained in negative and positive MS modes were compared. In positive MS mode only a few components could be detected (5 from the 15), which is understandable regarding the acidic feature of the components studied. In contrast, 4-(dimethylamino)benzoic acid contains an easily ionizable amino group, due to which considerably higher detection sensitivity could be observed.

Based on the measurements described above, the optimal parameters for the CZE-MS analysis of the studied components were 0.5 M NH_4_OH solution as BGE, pure IPA:water as sheath liquid and negative mode MS detection. Regardless of the utmost simplicity of the BGE and sheath liquid composition we used, their utilization for the CZE-MS analysis of phenolic compounds has not been reported in the literature so far. The only concern in connection with our BGE of choice is that components might be decomposed at such high pH, which was found in the case of quercetin. Apart from quercetin, all the components studied were stable during the analyses.

### 2.2. Analytical Performance Study

The analytical performance of the CZE-MS method developed was evaluated for the proposed separation and detection conditions of the 15 phenolic compounds studied. The calibration diagrams showed good linearities (better than 0.996 for the majority of components) in the concentration ranges of 0.1–200 mg/L and 2–500 mg/L for 5 and 9 components, respectively (Appendix A). The precision of the analyses was studied by the consecutive measurement of the mixture of standard solutions (Appendix A). The RSD data were better than 0.75% for migration times and ranged between 5–8% for peak areas. The relatively poor peak area precision can be attributed to the inaccurate integration of overlapped and slightly tailed peaks. The LOD values scattered between 0.004–1.9 mg/L, which indicates that there are relatively large differences in the ionization (and chemical) characteristics of these compounds. It is commonly known that the detection sensitivity of CZE methods can be largely enhanced by increasing the sample injection volume (for instance, the application of 60 s injection time instead of 6 s could lead to a seven-fold improvement of LOD values). The analytical performance data are summarized in Table 1.

In order to test the CZE-MS method developed, two extracts of a sunflower honey sample were analyzed in triplicate (Figure 4). Reversed phase SPE cartridge was used for sample clean-up, whereby the high glucose/fructose content was removed and the phenolic compounds were collected for subsequent analyses. Two different extractions were carried out: ACN:water = 7:3 + 0.1% FA and MeOH:ACN = 2:1. The peak patterns of the base peak electropherograms obtained for the two extracts showed similarities (several components in similar migration windows and some peaks with the same migration time appeared (e.g., peaks at 6.42, 7.95, 8.35 and 8.86 min)) as well as considerable differences (e.g., a high intensity component can be seen at 9.4 min in Figure 4b, while in Figure 4a it shows very low signal intensity).

The theoretical (calculated) and the experimentally obtained masses typically agreed within 1.5 ppm accuracy. Most masses (*m*/*z*) from among the components studied (12 from the 15) were detected in the extracts; however, in several cases (e.g., 3,4-dimethoxycinnamic acid, camphoric acid, salicylic acid, chrysin) there was a mismatch in migration times between the standard and the sample component of equivalent *m*/*z* value. Since the amount of detected components was often very low, the peak with the expected *m*/*z* could be observed only in the extracted ion electropherogram recorded at the proper *m*/*z* value.

Generally, the identification of analytes should be carried out by the comparison of their migration times and high-resolution mass spectrometry data with those of the standard solution. However, these data could not be matched for several peaks detected in the honey samples (e.g., peaks at 9.68, 9.95, 10.43 and 14.63 min). The explanation could be the presence of miscellaneous isomers. Although high-resolution MS provides the chemical formula for a given peak at the electropherogram with high probability through accurate mass measurement, a given chemical formula (elemental composition) can specify several possible chemical structures, and constitutional isomers. This means that the detected peak with an *m*/*z* value may belong to an isomer of the expected compound. In some works, NMR (besides the HPLC–Q-Exactive-Orbitrap^®^–MS analysis) was additionally used for analysis, especially when isomeric compounds had to be identified [25].

Several works can be found in the literature where the identification of a compound was based solely on the molecular mass detected in a natural sample. However, our results show that the use of HR-MS exclusively is not sufficient for high probability identification. In such cases, tandem MS might bring some insights into structural characteristics; however, the acquisition of valuable fragment mass spectra can be difficult for the small intensity peaks (components in real samples are often present only in a very low concentration). For instance, four different cinnamic acid isomers (C_9_H_8_O_2_) in propolis extract detected by HPLC-MS appeared in the chromatogram at very different retention times (5.87, 13.12, 13.81 and 16.15 min) [25]. In our study, only three phenolic components could be clearly identified based on matching with theoretical exact masses, migration times with standards and standard addition (spiking). There were 11 further components where only the isomer(s) could be determined, not the available and measured standard. Therefore, only three phenolic components could be quantitatively determined. For the qualitative and quantitative determination of the separated components in natural samples, a more detailed investigation would be necessary with proper standards or special, indirect analytical methods, tandem MS or NMR studies.

## 3. Materials and Methods

### 3.1. Reagents and Solutions

Analytical grade reagents were used. The phenolic compounds like 3,4-dimethoxycinnamic acid, chrysin, 4-(dimethylamino)-benzoic acid, cinnamic acid, benzoic acid, salicylic acid, hesperetin, naringenin, camphoric acid, ferulic acid, caffeic acid, o-coumaric acid, vanillic acid, p-coumaric acid and protocatechuic acid were obtained from Sigma (St. Louis, MO, USA). The standards were dissolved in methanol and the stock solutions were stored at 4 °C for max. 4 weeks. NH_4_OH and NH_4_Ac stock solutions (all Sigma products) were prepared in double-deionized water (Elix-3, Millipore, Darmstadt, Germany). HCl, NaOH, acetic acid (HAc) and formic acid (FA) solutions, isopropyl alcohol (IPA), methanol and acetonitrile were purchased from VWR (Radnor, PA, USA).

### 3.2. Samples and Its Pretreatment

The sunflower honey sample was obtained from a local producer. The honey was diluted with 10 mM HCl and the sugar content along with some other matrix materials was removed with the use of SPE (Dionex OnGuard RP cartridge). Two SPE procedures were applied according to ref. [26]: (1) Preconditioning the cartridge with 9 mL MeOH, 9 mL water and 9 mL 10 mM HCl, loading the sample (8.9 g honey diluted in 22.25 mL 10 mM HCl), washing with 12 mL 10 mM HCl, elution with 400 μL ACN:water = 7:3 + 0.1% FA and postconditioning with 9 mL ACN and 9 mL MeOH. (2) Preconditioning the cartridge with 5 mL MeOH; 5 mL water and 5 mL 10 mM HCl, loading the sample (6 g honey diluted in 9 mL 10 mM HCl), washing with 5 mL 10 mM HCl and 5 mL water, elution with 250 μL MeOH:ACN = 2:1 and postconditioning with 5 mL ACN and 5 mL MeOH.

### 3.3. Instrumentation

CZE-MS measurements were performed with a 7100 model CE instrument (Agilent, Waldbronn, Germany) coupled to a maXis II UHR ESI-QTOF MS (Bruker, Bremen, Germany) via a CE-ESI Sprayer interface (G1607B, Agilent). Sheath liquid was delivered with a 1260 Infinity II isocratic pump (Agilent). The CE instrument and the pump were controlled by OpenLAB CDS Chemstation software (Agilent). The MS was operated by otofControl version 4.1 (build: 3.5, Bruker) and the obtained electropherograms and mass spectra were processed by Compass DataAnalysis version 4.4 (build: 200.55.2969, Bruker).

### 3.4. CZE-MS Measurements

For CZE-MS analyses 90 cm × 50 µm id. fused silica capillaries were used. The BGE was 0.5 M NH_4_OH, the applied voltage was +30 kV and sample injection was performed with 50 mbar for 4 s. The postconditioning step involved washing with acetonitrile (4 bar, 2 min), water (4.5 bar, 2 min) and BGE (4.5 bar, 3 min). Sheath liquid consisted of IPA:water = 1:1 and was delivered at a flow rate of 6 µL/min to establish electric connection and stable electrospray formation. In the majority of cases the following parameters were applied for MS acquisition: negative polarity mode; nebulizer pressure: 0.4 bar; dry gas temperature: 200 °C; dry gas flow rate: 4 L/min; capillary voltage: 4500 V; end plate offset: 500 V; MS spectra rate: 2 Hz; mass range: 80–1000 *m*/*z*. Na–acetate adducts enabled internal *m*/*z* calibration.

## 4. Conclusions

In our study, a CZE-MS method was developed for the determination of several phenolic compounds (phenolic acids, flavonoids). As we ascertained that the analysis of these components requires CZE separation under basic conditions and negative ionization mode for MS detection, the simplest choice was to apply 0.5 M NH_4_OH and IPA:water (1:1 *v*/*v*%) for the BGE and the sheath liquid, respectively. The absence of acetate, formate or any salts (that are commonly used) provided a simple mass spectrum during the analyses. Proper precision of the measurements could be achieved even with these solutions having no buffer capacity, if frequent (at least after three runs) BGE replenishment and well-closed solution containers were used.

CE is considered suitable for the analysis of samples having high matrix content, since there are no considerable interferences arising during the separation. On the other hand, such minute (nanoliter ranged) amounts of sample are actually not expected to cause interferences or memory effects in the MS detection. Extracts of honey as a natural sample with many different types of compounds were used to test the developed CZE-MS method. The suggested SPE method required much lower amounts of samples and allowed easier extraction than others found in literature. In these measurements the unambiguous determination of several compounds was rather challenging, since the migration times and high-resolution mass spectrometry data obtained from the standard and the sample did not match. The high number of isomers, aglycones, glycosides or methylated derivatives of the phenolic compounds possibly occurring in natural samples obligates the use of proper standards and high-performance analytical approaches (e.g., tandem MS or NMR). Although the accurate molecular masses of several expected compounds were detected with high-resolution MS, the reliable qualitative and quantitative determinations were accomplished only for three components.

## Figures and Tables

**Figure 1 molecules-27-04540-f001:**
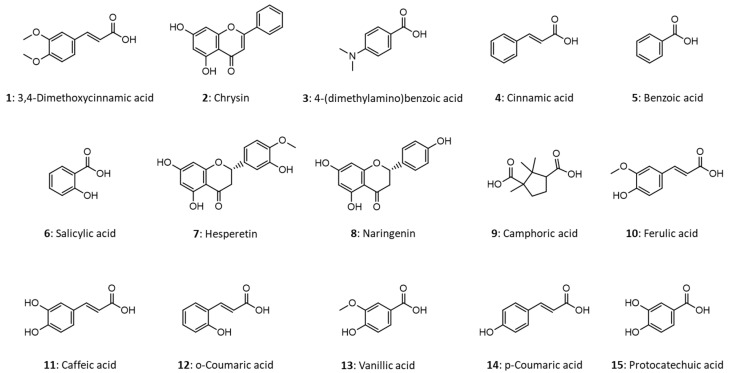
Molecular structures of the phenolic compounds studied.

**Figure 2 molecules-27-04540-f002:**
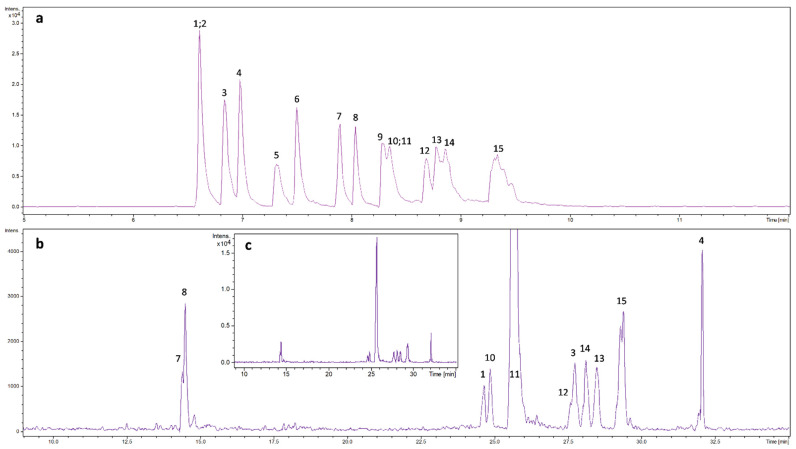
CZE-MS electropherograms obtained with (**a**) 0.5 M NH_4_OH (pH = 11.4) and (**b**) 50 mM ammonium-acetate (pH = 7) including an inset (**c**) which represents (**b**) the electropherogram with a larger intensity scale. The number of the peaks correspond with the number assigned to the components given in Figure 1. Conditions: capillary: 90 cm × 50 µm, U = +30 kV, sample injection: 50 mbar × 4 s, MS: negative ionization mode, sheath liquid: IPA:water = 1:1, sheath liquid flow: 6 μL/min, nebulizer pressure: 0.8 bar, drying temperature: 200 °C, spectra rate: 2 Hz. Sample: 3.833 mg/L 3,4-dimethoxycinnamic acid (1), 0.209 mg/L chrysin (2), 11.63 mg/L 4-(dimethylamino)benzoic acid (3), 17.44 mg/L cinnamic acid (4), 49.87 mg/L benzoic acid (5), 22.89 mg/L salicylic acid (6), 1.765 mg/L hesperetin (7), 1.637 mg/L naringenin (8), 7.881 mg/L camphoric acid (9), 4.968 mg/L ferulic acid (10), 85.66 mg/L caffeic acid (11), 19.98 mg/L o-coumaric acid (12), 15.69 mg/L vanillic acid (13), 35.706 mg/L p-coumaric acid (14), 8.73 mg/L protocatechuic acid (15).

**Figure 3 molecules-27-04540-f003:**
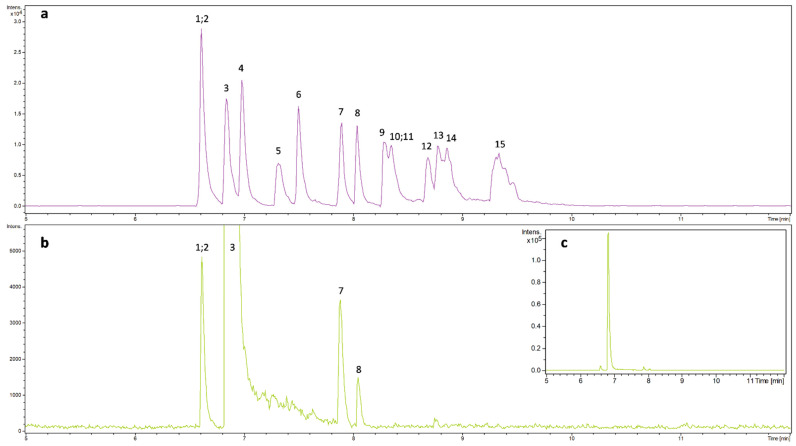
CZE-MS electropherograms obtained using 0.5 M NH_4_OH as BGE with (**a**) negative and (**b**) positive ionization mode. The inset (**c**) represents the electropherogram with a larger intensity scale in positive ionization mode. Conditions are the same as in Figure 2a, but for (**b**,**c**) the sheath liquid was 0.1% formic acid in IPA:water = 1:1 and the positive ionization mode was applied.

**Figure 4 molecules-27-04540-f004:**
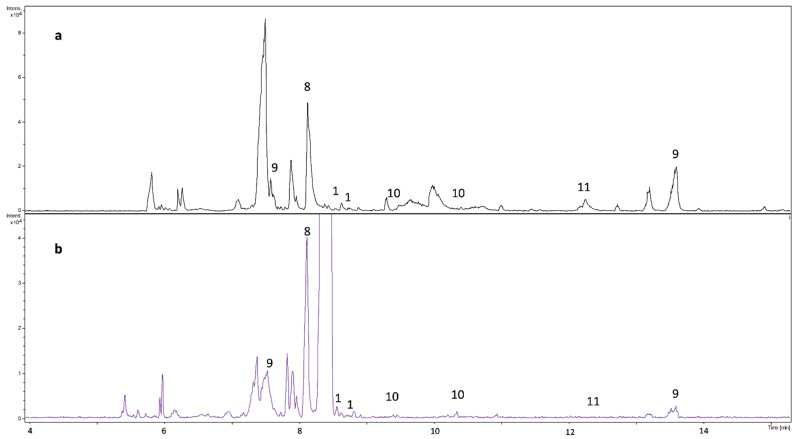
Base peak CZE-MS electropherograms obtained for two different SPE extracts of the same honey sample (sunflower). Conditions are the same as in Figure 2a. The number of the peaks correspond with the number assigned to the components given in Figure 1. SPE (Dionex OnGuard RP cartridge) for (**a**): preconditioning the cartridge with 9 mL MeOH, 9 mL water and 9 mL 10 mM HCl, loading the sample (8.9 g honey diluted in 22.25 mL 10 mM HCl), washing with 12 mL 10 mM HCl, elution with ACN:water = 7:3 + 0.1% FA and postconditioning with 9 mL ACN and 9 mL MeOH. SPE for (**b**): preconditioning the cartridge with 5 mL MeOH; 5 mL water and 5 mL 10 mM HCl, loading the sample (6 g honey diluted in 9 mL 10 mM HCl), washing with 5 mL 10 mM HCl and 5 mL water, elution with MeOH:ACN = 2:1 and postconditioning with 5 mL ACN and 5 mL MeOH.

**Table 1 molecules-27-04540-t001:** Analytical Performance Data of CZE-MS Measurements.

#	Name	Formula	[M-H]^−^ Mass	Equation for Calibration Graph	R^2^	Linear Range (mg/L)	LOD (mg/L)	Recovery(%) ^1^	Conc. in Honey (mg/kg) ^2^	RSD% (Min)	RSD% (Area)
1	3,4-Dimethoxycinnamic acid	C_11_H_12_O_4_	207.0663	y = 14.86x − 35.19	0.9942	0.1–200	0.026	88.2		0.59	7.69
2	Chrysin	C_15_H_10_O_4_	253.0506	y = 36.23x − 8.07	0.9948	0.1–200	0.004	84.6		0.68	6.9
3	4-(dimethylamino)-benzoic acid	C_9_H_11_NO_2_	164.0717	y = 5.16x − 0.24	0.9944	0.5–500	0.067	86.1		0.63	7.42
4	Cinnamic acid	C_9_H_8_O_2_	147.0452	y = 3.49x − 9.18	0.9995	0.5–500	0.153	94.2		0.65	6.37
5	Benzoic acid	C_7_H_6_O_2_	121.0295	y = 0.54x + 13.7	0.9982	10–1000	0.628	95.6		0.65	5.27
6	Salicylic acid	C_7_H_6_O_3_	137.0244	y = 3.49x − 37.0	0.9966	2–500	0.323	84.9		0.56	6.61
7	Hesperetin	C_16_H_14_O_6_	301.0718	y = 28.6x − 6.17	0.9996	0.1–200	0.016	84.8		0.49	5.93
8	Naringenin	C_15_H_12_O_5_	271.0612	y = 24.3x + 11.0	0.9992	0.1–200	0.011	95.7	0.599 ± 0.036	0.42	5.62
9	Camphoric acid	C_10_H_16_O_4_	199.0976	y = 7.45x + 19.4	0.9998	0.5–500	0.079	91.2		0.38	6.35
10	Ferulic acid	C_10_H_10_O_4_	193.0506	y = 11.1x + 14.1	0.9990	0.1–200	0.06	96.4	0.0130 ± 0.001	0.47	5.18
11	Caffeic acid	C_9_H_8_O_4_	179.035	y = 1.13x − 20.0	0.9860	10–500	1.921	95.9	3.369 ± 0.22	0.75	7.74
12	o-Coumaric acid	C_9_H_8_O_3_	163.0401	y = 4.58x − 7.42	0.9824	2–500	0.517	87.3		0.17	6.06
13	Vanilic acid	C_8_H_8_O_4_	167.035	y = 3.41x + 10.4	0.9995	2–500	0.197	86.2		0.13	6.28
14	p-Coumaric acid	C_9_H_8_O_3_	163.0401	y =15.1x + 4.73	0.9886	0.5–500	0.085	97.0		0.28	7.31
15	Protocatechuic acid	C_7_H_6_O_4_	153.0193	y = 4.31x − 38.0	0.9995	2–500	0.33	99.4		0.73	6.42

^1^ sample solution was spiked with 6.67 mg/L standards. ^2^ SPE was performed by an elution with ACN:water = 7:3 + 0.1% FA.

## Data Availability

The data presented in this study are available on request from the corresponding author.

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
