# Peer review of "Determination of Phenolic Compounds by Capillary Zone Electrophoresis–Mass Spectrometry"

_molecules, 2022, doi:10.3390/molecules27144540_

Round 1

Reviewer 1 Report

The authors have presented their approach to conducting the analysis of phenols in honey via CE-MS and have successfully separated and identified the compounds of interest.  However, I am struggling to find substantive differences between this work and the work conducted by D. Arráez-Romány et al. https://doi.org/10.1016/j.jpba.2006.02.035 (reference #20 in this manuscript).  The authors need to clearly distinguish how their work can be differentiated from the prior publication.  At present both this manuscript and the 2006 paper use SPE-CE-MS to analyze phenols in honey.  Both use a basic ammonium buffer (the only difference being the anion in the buffer: acetate vs. hydroxide) with isopropanol as an additive. It is unclear to me what innovation or advancement has been made by the authors in this work to justify its publication.

The study has been well executed, and should the authors be able to clearly articulate the substantive difference in this work to that of Arráez-Romány et al. I believe that this manuscript would be acceptable for publication.

Reviewer 2 Report

The authors report an electrophoretic method with MS detection for the determination of several phenolic compounds.

The manuscript is well written and structured, but a deep revision is necessary to correct some misspellings and typos.

The developed EC method seems to be easy to implement and efficient for the resolution of 15 compounds and I have no comments about CE methodology, however, some important issues must be explained and experiments are missing and, in consequence, the manuscript cannot be published in this way.

Minor issues:

I think the title must be a little more informative

CZE must be clarified first in the text before using it.

Major issues:

In the methodology, something that catches my attention is the high concentration levels of the calibration curves. Are these compounds found at such high concentration levels in natural samples?

There is no recovery analysis performed. So, the analytical method cannot be evaluated. Moreover, there are no analytical figures evaluated such as repeatability, reproducibility, and so. Then, the analytical performance cannot be established in any way.  

More analysis should be first carried out in order to evaluate the performance of the method and, then, its applicability. Recoveries in validation samples need to be performed. Then, in the real natural samples. Otherwise, the concentrations found in honey samples lack reliability. At last, pre-concentration, if there is some, should be calculated.

This is the most important deficiency of the manuscript. Without a proper analytical performance analysis, it cannot be published.

For all these, I cannot suggest the manuscript for further publication.

Round 2

Reviewer 1 Report

I appreciate that the authors have clarified the differences between their work and that of D. Arráez-Romány et al. the differentiation was necessary for understanding the context of their work.

When I first reviewed this work I was rather focused on the lack of differentiation with prior work, and overlooked a few items that will need to be addressed before the manuscript can be sent for publication.

Major Issues

Peak identification in figures and text.

The revision of the manuscript did not include the figures, which I presume is merely an oversight, however in the first draft of the manuscript the authors provided very little annotation of the figures, making it difficult to interpret both the results of the separations and the author’s conclusions.  Specifically, only the captions for figure 2 list the compounds tested, but only figure 3 has notations identifying peaks (though these numbers are not clearly associated with analytes, the reader needs to count the compounds in the list).  The authors need to do a better job of annotating the peaks in their electropherograms so that the reader can properly interpret the data.

In addition, the authors are overly vague in their description of the data that they have observed.  This is most notable in the section of the Analytical Performance Study covered in lines 152 - 191.  The language such as “several components in similar migration windows and some peaks with the same migration time appeared” does not aid the reader in interpreting if the methods are of utility for their needs. Please take the time/space to list the compounds that are being discussed.

Quantification data in Table 1

In table 1 there is a column labelled Conc. in honey (mg/kg) with data for naringenin, ferulic acid, and caffeic acid.  The authors however do not provide any information on how these concentrations were acquired.  Presumably this is through the method that they are presenting, but they have presented two methods for sample preparation, each seaming to yield different results, so clarification on which was used is important.

The authors should also report the standard deviation in the reported concentrations of the compounds found in the honey.

Additionally, it is concerning that the reported concentration of ferulic acid is roughly five times lower than the authors reported LOD.  The authors should explain how they were able to achieve this measurement.

Finally, and this may related to my questions about the quantification of the ferulic acid, how does the LOD, in mg/L, translated into the mg/kg concentration reported for the honey?  

Supplementary Information

The authors provided supplementary information in response to Reviewer 2’s questions about the recovery of their method.  However, the authors do not make any mention of the recovery in the updated manuscript, nor do they point the reader to the existence, or utility of the supplemental information.  In addition the authors have not specified which of their two methods was being evaluated through these recovery measurements, or if the results are applicable to both methods.

Reviewer 2 Report

The authors respond to all the enquiries. 

Author Response

We thank the reviewer for her/his work.